# Identification of Cellulose-Degrading Bacteria and Assessment of Their Potential Value for the Production of Bioethanol from Coconut Oil Cake Waste

**DOI:** 10.3390/microorganisms12020240

**Published:** 2024-01-24

**Authors:** Zihuan Fu, Longbin Zhong, Yan Tian, Xinpeng Bai, Jing Liu

**Affiliations:** 1Engineering Research Center of Utilization of Tropical Polysaccharide Resources, Ministry of Education, Hainan University, Haikou 570228, China; 20086000210063@hainan.edu.cn (Z.F.); 810045@hainanu.edu.cn (L.Z.);; 2International School of Public Health and One Health, Hainan Medical University, Haikou 571199, China

**Keywords:** bioethanol, agro-industrial waste, cellulose, *Bacillus*, cellulose-degrading bacteria, coconut oil cake

## Abstract

Bioconversion of lignocellulosic biomass is a highly promising alternative to rapidly reduce reliance on fossil fuels and greenhouse gas emissions. However, the use of lignocellulosic biomass is limited by the challenges of efficient degradation strategies. Given this need, *Bacillus tropicus* (*B. tropicus*) with cellulose degradation ability was isolated and screened from rotten dahlia. The strain efficiently utilized coconut oil cake (COC) to secrete 167.3 U/mL of cellulase activity. Electron microscopy results showed significant changes in the structure and properties of cellulose after treatment with *B. tropicus*, which increased the surface accessibility and the efficiency of the hydrolysis process. The functional group modification observed by Fourier transform infrared spectroscopy indicated the successful depolymerization of COC. The X-ray diffraction pattern showed that the crystallinity index increased from 44.8% to 48.2% due to the hydrolysis of the amorphous region in COC. The results of colorimetry also reveal an efficient hydrolysis process. A co-culture of *B. tropicus* and *Saccharomyces cerevisiae* was used to produce ethanol from COC waste, and the maximum ethanol yield was 4.2 g/L. The results of this work show that *B. tropicus* can be used to prepare biotechnology value-added products such as biofuels from lignocellulosic biomass, suggesting promising utility in biotechnology applications.

## 1. Introduction

The demands of social development and the surge in population have placed an enormous burden on resources and their use, leading to an increase in total global energy consumption of about 200% [1]. The depletion of fossil fuel resources and the increase in greenhouse gas emissions have led to energy crises and environmental pollution, prompting the development of green and renewable energy sources [2,3]. The production of biofuels from lignocellulosic biomass (e.g., agricultural residues and industrial wastes) is of global interest as biomass represents a large renewable energy source that could effectively meet the current challenges of global energy demand [4,5,6,7]. Globally, bioethanol is the most used liquid biofuel (about 80%), and its use not only reduces greenhouse gas emissions but also helps to reduce the generation of particulate matter in the atmosphere [1]. Bioethanol is now widely accepted in the United States, Brazil, the European Union, and China [8]. In China, 7.5 × 10^8^ tons of lignocellulose are available annually, which could theoretically produce 40–50 million tons of bioethanol per year [9,10]. Converting industrial and agricultural wastes into biofuels through sugar would significantly lessen the energy crisis and alleviate environmental pollution, which is of great significance to sustainable development.

COC, a significant solid waste from the coconut oil industry, is low-cost, renewable, and readily available [6]. COC has been used as an ingredient in low-cost animal feeds due to its significant amount of protein (20–25%), but the high cellulose content of COC and low lysine concentration have led to a global decrease in its use in animal feed formulations. Thus, a large amount of COC is burned for disposal, resulting in environmental pollution [6,11,12,13]. In tropical areas, coconut is widely available and serves as the main energy crop. Its heterogeneous biomass contains lignocellulose (hydrophilic biomass) and waste coconut oil (hydrophobic biomass) and is used to produce various valuable products, including raw materials for biofuels. Many previous studies have investigated the biodiesel production of COC substrates [6,14]. COC is a natural biological source of carboxymethyl cellulose, explaining the maximum cellulase yield [6,14]. Given the abundance of this waste resource and its potential to make biofuel, studying cellulase-mediated biodegradation of COC as a feedstock for bioethanol production is a promising approach to sustainability [15].

Lignin, cellulose, and hemicellulose can be converted to monosaccharides in COC, with lignin adhering to the outermost layer for easy removal and hemicellulose having a loose structure for easy degradation. Cellulose is a serious obstacle to the first step of bioethanol production due to its water-insoluble and highly crystalline structure, so the use of COC in bioethanol production requires a strategy for the effective degradation of cellulose. In the global carbon cycle, carbohydrate-active enzymes in microorganisms can depolymerize complex lignocellulosic polysaccharides. Thus, applying the potential of microorganisms for the hydrolysis of cellulosic biomass may be an eco-friendly method with a low yield of toxic substances and mild reaction conditions [16]. In the past, filamentous fungi have been widely and intensively studied for their efficient cellulose degradation properties. However, cultures of fungi are difficult to expand, exhibit high environmental sensitivity, and may not be suitable for large-scale industrial production [17]. As an alternative to fungi, bacteria are now being explored for use in the enzymatic hydrolysis of lignocellulose due to their high rate of enzyme production, expression of multi-enzyme complexes, tolerance to extreme environments, and the ability to be genetically engineered [5,18]. Previous studies have reported the isolation of cellulose-degrading bacteria from nature to modify the recalcitrant structure of cellulose to add value [19]. Therefore, the screening and identification of high-efficiency cellulolytic bacteria can provide insight into cellulose degradation mechanisms and inspire methods to improve the utilization rate of cellulose resources.

The goal of this study was to isolate and identify cellulose-degrading bacteria from decaying plants and screen the isolated bacteria for their ability to hydrolyze cellulose. The ability of the isolated bacteria to biodegrade COC was then analyzed, and the cellulose degradation mechanism of the isolated strain was determined by observing the microstructure, chemical structural properties, pyrolysis characteristics, and apparent color of COC during the degradation process. Finally, the screened bacteria and *Saccharomyces cerevisiae* were used for COC fermentation to bioethanol. This study demonstrates the feasibility of finding natural bacterial isolates that can promote biofuel production and the rational utilization of waste COC resources.

## 2. Materials and Methods

COC was provided by the Coconut Oil Processing Factory in Haikou, Hainan, China. The collected COC was crushed into powder form using a crusher (Tianjin Meister Instruments Co., Ltd., Tianjin, China). Samples of rotten dahlia plants were collected from Hainan Botanical Garden in Danzhou, China (the required permission was obtained for the use of this material). Bacterial genomic DNA isolation kits were obtained from Nanomagnetic Biotechnology Ltd. (Wuhan, China). Ethanol and 3,5-dinitrosalicylic acid reagents were obtained from McLean Biochemicals Ltd. (Lanzhou, China). All reagents were 95–98% analytically pure.

### 2.1. Isolation of Cellulolytic Bacteria

Highly decayed dahlia plants were collected from Hainan Botanical Garden for screening of cellulose-degrading bacteria. The method of Duhan et al. [18] was selected for adjustment. The heavily decayed plant tissues were taken with sterile forceps and torn apart. About 1 cm of the above tissue samples was immersed in a 10% sodium bicarbonate solution (10 min) to destroy and inhibit fungal growth. The samples were aseptically dried, placed on nutrient broth agar media, incubated at 37 °C for colony growth, and plated repeatedly to obtain isolated bacteria.

### 2.2. Qualitative Screening

Congo red staining was used to evaluate the ability of the isolated bacterial strains to hydrolyze cellulose [16]. To perform this, bacterial isolates were applied to CMC-Na agar and incubated at 37 °C for 3 days. At the end of the incubation period, the agar plate was stained with Congo red aqueous solution (2 g/L) for 20 min, and then the dye was removed by rinsing with NaCl solution at a concentration of 1 mol/L. The formation of transparent zones indicated areas of cellulose degradation [5]. Finally, the diameter D of the hyaline zone produced by the bacteria and the diameter d of the colony were measured with vernier calipers, and the experimental samples were compared. Those with larger ratios were selected for additional testing.

### 2.3. Quantitative Screening

Quantitative screening was performed to compare the degradation activities of different bacteria by measuring cellulase (CMCase) activity. Each bacterial isolate was separately grown in a 150 mL conical flask containing 50 mL of medium with CMC-Na (10 g/L) as the sole carbon source and incubated for 3 days at 37 °C [16]. CMCase was quantified with reference to the 3,5-dinitrosalicylic acid reagent as described by Kledson et al. [18], with some modifications. The standard curve was constructed by measuring the OD 520 (Synergy LX, Blotek, CA, USA) value using a 1 mg/mL glucose standard solution as the substrate. In each reaction, 500 μL of crude enzyme solution and 1500 μL of 1% (*w*/*v*) CMC-Na (pH 5.5) were mixed and incubated at 45 °C for 35 min. To this mixture, 1500 μL of 3,5-dinitrosalicylic acid reagent was added for coloration, and the enzymatic reaction was terminated by incubation in boiling water for 13 min. Then, 3.5 mL of the mixture was cooled with running water for 5 min, and the absorbance value was measured at 520 nm. The unit enzyme activity was defined as the amount of enzyme required to release 1 µmol of reducing sugar (calculated as glucose) per minute during the reaction.
$$\mathrm{enzyme}\;\mathrm{activity}=\frac{F-f}{30}$$
*F*: blank glucose amount (μg);*f*: Glucose weight of the sample (μg);30: The reaction time (min) between the enzyme and the substrate.


### 2.4. Fermentation Enzyme Production Curve

To evaluate the ability of the screened strains to degrade COC, 10 mL of freshly prepared bacterial culture was inoculated into a culture medium with COC as the sole carbon source [20]. Samples were collected every 12 h, and the degradation efficiency was determined according to the method described in Section 2.3.

### 2.5. Bacterial Identification and Phylogenetic Analysis

Genomic DNA was isolated using a bacterial genome extraction kit, and the 16S rRNA fragment was amplified by Polymerase Chain Reaction. The amplified products were detected on a 1% agarose gel and sent to Hainan Liffet Biotechnology Co., Ltd., Haikou, China, for sequencing. MEGA X software 10.1.8 (Mega Limited, Auckland, New Zealand) was used to compare the sequences, and the neighbor-joining (NJ) method was used to construct the phylogenetic tree.

### 2.6. Field Emission Scanning Electron Microscopy (FESEM)

FESEM was used to evaluate the microstructural changes of COC before and after bacterial treatment. Briefly, representative micrographs of the samples were taken at 5 kV using a S-4800 FESEM (Thermo Scientific, Waltham, MA, USA) [21].

### 2.7. Fourier-Transformed Infrared (FTIR)

FTIR was used to characterize and determine the functional groups of COC before and after bacterial treatment. The procedure was as follows: TENSOR 27 FTIR (Bruker Company, Bremen, Germany) was used to mix and grind a small amount of dried sample with KBr before pressing the material into tablets [22]. FTIR was performed with an experimental wavelength measurement range of 500–4000 cm^−1^, 32 scans, and a resolution of 2 cm^−1^. Before collecting data, background signals were measured and used for correction.

### 2.8. X-ray Diffraction (XRD)

XRD is commonly used to evaluate physicochemical indicators such as cellulose crystallinity. A Smart Lab X-ray diffractometer (Rigaku, Woodlands, TX, USA) was used to measure COC before and after degradation. For scanning, the 2θ angle was within the range of 5~40°, with a scan step of 0.02° and a scan rate of 2.0°/min. Based on the intensity of the diffraction peaks, the crystallinity index (CrI) was calculated based on the “Segal method” [23]:$$\mathrm{CrI}=\frac{I_{002}-I_{am}}{I_{002}}\times 100$$
where *I*_002_ represents the intensity of the peak at approximately 2θ = 22, and *I_am_* represents the peak intensity in the non-crystalline region at approximately 2θ = 18 [24].

### 2.9. Color Analysis

The color change of COC was measured at each stage according to Wang et al. [25], with slight modifications. Briefly, the color of the samples was determined using an SPH 860 colorimeter (Colorlite, Katlenburg-Lindau, Germany). COC color was evaluated from three aspects using the CIE Lab color system, where an appropriate amount of dry powder was placed on the sample plate to directly measure the color L*, a* and b* coordinates [19]. Each value was measured nine times, and the average value was taken.

### 2.10. Enzymatic Hydrolysis and Ethanol Production

A bacterial culture was used to inoculate medium containing 20 g/L COC as substrate in a 100 mL conical flask. The inoculated conical flasks were incubated at 37 °C with shaking at 125 rpm (Aohua, Changzhou, China). Samples were taken out every 12 h and centrifuged at 8000 rpm for 6 min (Tianjin Meist Instrument Co., Ltd., Tianjin, China). The supernatant and the released reducing sugars were determined using the DNS method [5]. Ethanol production was also investigated. First, the hydrolysis products were inoculated with 10% (*v*/*v*) *Saccharomyces cerevisiae* seed cultures (freshly cultured for 72 h), and 1 mL samples were removed every 12 h and centrifuged at 8500 rpm for 6 min. Finally, K_2_Cr_2_O_7_ reagent was used to evaluate the ethanol production of the resulting supernatant [20]. The ethanol produced was quantitatively estimated at 600 nm using a multi-function microplate reader (Varioskan LUX, Thermo Fisher, Waltham, MA, USA).

## 3. Results and Discussion

### 3.1. Screening of Cellulose-Degrading Bacteria

Cellulose hydrolases are naturally produced by microorganisms and can convert cellulosic biomass into fermentable sugars [3]. There is a great demand for cellulose-degrading microorganisms in industrial applications, so searching for biomass-degrading bacteria from different environments is a key research focus [26]. In this work, we aimed to isolate cellulolytic bacteria from environmental systems to degrade COC. A total of 20 strains of bacteria were isolated and purified from decayed dahlias by selecting and purifying different colonies in successive streaks on agar media. CMC-Na agar medium was used to cultivate microorganisms, monitor their growth, and evaluate the ability of pure bacterial strains to secrete cellulase by the Congo red method. Of the 20 strains, eight isolates expressing cellulase activity were selected, as shown in Figure 1a. As measured by the Congo red method, the cellulose hydrolysis capacities of BD1, BD2, BD10, BD15, BD16, BD17, BD19, and BD20 strains were 3.02, 1.51, 3.815, 1.705, 1.215, 2.59, 3.19, and 2.61 cm, respectively. Among the eight strains that produced hyaline rings, the hydrolysis capacity of BD10 was higher, suggesting that this strain produces higher cellulase enzyme activity than the new isolates. However, hyaline ring measurement cannot fully represent the enzyme-producing ability of the strain because the size of the cellulose-dissolving zone around a colony reflects both the enzyme activity and the characteristics of the strain itself. Thus, to more carefully investigate the hydrolysis capacity of these strains, the enzyme activity was quantitatively determined by spectrophotometry. As shown in Figure 1b, the CMCase activities of BD1, BD2, BD10, BD15, BD16, BD17, BD19, and BD20 were 67.68, 66.5, 170.12, 71.45, 45.34, 90.11, 122.87, and 75.12 U/mL, respectively.

BD10 showed the highest cellulase hydrolysis ability in the evaluation of primary screening (a qualitative experiment) and secondary screening (a quantitative experiment) (*p* < 0.0001) (Figure 1b). The cellulase hydrolysis ability of BD10 was 26.32, 152.65, 123.75, 213.99, 47.29, 19.59, and 46.17 higher than that of BD1, BD2, BD15, BD16, BD17, BD19, and BD20, respectively. For cellulase enzyme activity, BD10 was higher than BD1, BD2, BD15, BD16, BD17, BD19, and BD20 by 151.4, 155.82, 138.09, 275.2, 88.79, 38.45, and 126.46%, respectively. Therefore, BD10 was selected for additional study. Li et al. [27] selected a bacterial strain with high cellulase production by Congo red assay and characterized its cellulase production. Darwesh et al. [16] selected the fungus with the highest cellulase yield of nine fungal isolates by qualitative analysis of Congo red and quantitative analysis of CMCase activity and used the selected fungus to degrade and saccharify rice straw, release reducing sugar, and further convert it into bioethanol. Our results confirm that cellulase-producing bacteria can be isolated from a decaying environment for potential use in a biorefinery environment.

### 3.2. Determination of Enzyme Production Efficiency in the Degradation of COC

According to current estimates from FAOStat 2022, the global annual waste generation from all crops is 3.71 × 10^10^ kg, and agricultural waste production is expected to increase by more than 50% by 2050 [28]. Therefore, there is an urgent need for efficient and low-cost cellulolytic bacteria that can help to degrade this waste. There are limitations to the use of transient enzyme activity during the pre-screening of strains, as well as differences in the cellulase systems induced by different cellulose substrates, so the CMCase activity of fermenting COC by strains without pretreatment was measured in real-time to more accurately represent the COC degradation ability. The release of CMCase was measured with culturing at 37 °C and 125 r/min, as shown in Figure 2. The results show that the isolated strain can hydrolyze COC and produce relatively stable cellulase. Overall, the cellulase activity increased first and then decreased with the extension of fermentation time, with the highest enzyme production of 167.3 U/mL at 36 h. According to the data, the first 36 h were the latent period, and the enzyme activity gradually reached its maximum as the concentration of the bacterium increased and the substrate induced the accumulation of CMCase. Later, as the nutrients in the medium were depleted, the aging of the bacteria and the decrease in its enzyme production capacity led to reduced enzyme secretion. Similarly, Darwesh et al. [16] reported that *Aspergillus niger* exhibited steadily increasing CMCase activity before reaching optimal yield on day 7. The changing trend of CMCase activity reported by Ali et al. [2] is similar to that of this study. In the first 10 days of culture, the bacteria used cellulose as a carbon source for growth and metabolism, suggesting high cellulose degradation activity.

### 3.3. Molecular Biology of Strains

In the pre-screening assay, strain BD10 showed the highest cellulase catabolic activity and produced relatively stable CMCase in the presence of COC as a substrate. Thus, strain BD10 was selected for identification. The 16S rRNA gene sequence of BD10 was used to search the NCBI BLAST database, and the results revealed high similarity with *Bacillus tropicus*, suggesting that BD10 is a species of the genus *Bacillus*. The phylogenetic tree based on the sequence was constructed by the NJ method (Figure 3) and also shows that BD10 is located in the pedigree of Bacillus. Based on these results, the strain is hereafter designated as *B. tropicus*. Bacillus strains are aerobic cellulolytic species that are frequently isolated from soils and waste sites with high cellulose content [29]. Bacillus strains have unique advantages of high-temperature resistance, acid and alkali resistance, and UV resistance, making them promising candidates for applications in biotechnology such as cellulase production [8]. Dar et al. [29] reported that *Bacillus subtilis* Cf60 showed the highest cellulase activity on filter paper as a substrate. Jiaxing et al. [30] found that *Bacillus* BS-5 could produce a cost-effective lignocellulose-decomposing enzyme mixture. *B. tropicus* was previously isolated as a potential organic pollutant-degrading bacterium. Duhan et al. [31] isolated *B. tropicus* from the leaves and stems of the medicinal plant *Pteris officinale*, leading to the identification of new bioactive metabolites and the development of potential biological pesticides and fertilizers. Zhu et al. [32] isolated *B. tropicus* from activated sludge from wastewater treatment plants and demonstrated its ability to promote the biodegradation of pyridine and quinoline.

We compared the activity of the *B. tropicus* isolated in this study with the previously reported cellulase-producing activities of different *Bacillus* strains, as shown in Table 1. The observed differences may reflect variations in the cellulose hydrolysis activities of the strains, differences in the analysis conditions, or variations in the specificity of the substrates. COC biomass can clearly induce cellulase production, and *B. tropicus* effectively utilizes COC to secrete cellulase, suggesting this strain may be feasible for the microbial hydrolysis of agricultural wastes.

### 3.4. Characterization of COC before and after Degradation

#### 3.4.1. FESEM Analysis

The characterization of the hydrolyzed substrate can provide insight into the mechanism of degradation. To perform this, *B. tropicus* treatment effects on COC were assessed by FESEM. FESEM-based visualization was performed to compare the surface morphology of the control and treated COC (Figure 4). The images of the untreated samples (Figure 4a,b) show a solid surface with a rigid and firm structure compared with the treated samples, with good integrity, a smooth and dense surface, tight connections between components, and few grooves or depressions. This ordered and tight structure makes it difficult for cellulase to access and degrade cellulose. The action of extracellular enzymes and small-molecule compounds secreted by *B. tropicus* caused significant morphological changes in the surface structure of COC. Images of COC treated with *B. tropicus* for 5–10 days (Figure 4c–f) revealed a rougher surface with the substrate cracked and delaminated, with pores and cracks in the lignocellulose, likely caused by the secretion of hydrolase by *B. tropicus.* This action of bacteria may affect the specific surface area of COC, increasing the accessibility of cellulase and facilitating the subsequent enzymatic digestion [41,42,43]. Many researchers have made similar observations of microbially degraded lignocellulosic biomass [44]. Dar et al. [45] concluded that the enzymatic hydrolysis of biodegradable polymers proceeds mainly through a surface erosion mechanism, where microorganisms initially engulf the external polymer of the substrate and then release toxic metabolites as esterases or organic acids. The released compounds can damage the surface of the polymer to expose the inner chains of the composite, increasing their susceptibility to enzymatic action and therefore accelerating degradation. In the late stage of fermentation (Figure 4g–j) with prolonged treatment, the material’s morphology shows deeper and more angular degradation grooves and increased pores. The material appears fluffy, indicating extensive hydrolysis of the fibrous fraction of the COC by cellulase, leading to the destruction of polysaccharide chains and an increase in the specific surface area [46,47]. These results are consistent with those of Liu et al. [48], who used SEM and observed that the outer surface of treated corn cobs became cracked and somewhat wrinkled due to the breakage of the complex lignocellulosic structure of the biomass when treated by fungi. Singh et al. [49] also found that microbial attack causes changes in the original fiber at the molecular level and determined that the formation of pores and cracks during degradation is correlated with the amount of reducing sugar produced.

#### 3.4.2. FTIR Analysis

FTIR was next used to examine the structure of COC biomass before and after bacterial treatment. The observed structural changes of COC reveal overall deformation and contraction of the fiber structure due to *B. tropicus* (Figure 5). In the data, the band at about 3410 cm^−1^ is attributed to the O-H stretching of hydrogen bound to hydroxyl groups, mainly from cellulose, and COC exhibits a decrease in peak intensity and width with enzymatic hydrolysis triggering cellulose degradation [50]. Bacterial action reduced the intensities of lignin absorption bands at 2852 cm^−1^ and 1743 cm^−1^, corresponding to C-H vibrations of methylene and carbonyl groups, respectively, verifying lignin side chain breakage [51]. There was also a decrease in absorption intensity near 1243 cm^−1^, indicating a decrease in hemicellulose/cellulose in the treated COC [5]. The enhancement of the spectral band at 898 cm^−1^ indicates the degradation of the amorphous component of the cellulosic material. The peak at 810.61 cm^−1^ corresponding to the stretching vibration of β-glycosidic bonds in polysaccharides, also decreases with *B. tropicus* treatment, suggesting that enzymatic hydrolysis can disrupt intermolecular interactions [43]. Our results are in agreement with those of Dar et al. [45], who observed a similar peak for cellulose at 897 cm^−1^. Overall, these findings demonstrate the effective utilization of the cellulose component of the COC sample by *B. tropicus*, which targets hydrogen bonds. The FTIR spectra are in agreement with the enzymatic assay and FESEM results, which together demonstrate the potential of COC as a substrate for biofuel production.

#### 3.4.3. XRD Analysis

Crystallinity is the primary determinant of cellulase hydrolysis [52]. XRD was employed to measure the crystallinity of the COC, including hemicellulose and lignin, as shown in Figure 6. All samples have regular broad peaks from 14.8° to 28°, which is characteristic of the crystalline form of cellulose, and typical diffraction peaks at around 16.09° and 22° [42,43]. After *B. tropicus* liquid fermentation, the peak pattern of the treated COC was consistent with that of untreated COC, indicating that *B. tropicus* did not completely disrupt the crystal structure of COC. The CrI of the treated material was increased compared to that of the control group without treatment [24]. This is because the substances secreted by *B. tropicus* preferentially remove the amorphous regions of the fiber structure (including hemicellulose and lignin), and the crystalline regions are resistant to attack by microorganisms and enzymes due to their dense structure. This resistance to degradation leads to an overall increase in the crystallinity index [45]. In particular, the disruption of β-1,4 bonds by CMCase leads to chain breaks and reduces the amorphous content to increase the crystallinity index. The degradation of amorphous biomass can expose more binding sites, thereby increasing enzyme accessibility to cellulose and improving enzymatic hydrolysis yields. Our results are also in agreement with those of Dar et al. [45], who similarly observed an increase in sawdust crystallinity after bacterial treatment. The XRD pattern obtained was consistent with the CMCase, FESEM, and FTIR results, reflecting the change of biomass functional groups and the increased exposure of cellulose due to bacterial hydrolysis.

#### 3.4.4. Color Analysis

During the fermentation period, the biological attachment can cause a color change in the substrate. Figure 7 shows that *B. tropicus* treatment caused color changes in the COC, with a significant color difference between the post-degradation samples and the pre-degradation samples. The L* value (brightness) decreased from 86.1 to 56.5, which suggests that oxidation of phenolic compounds or the formation of pigments occurs during cellulase hydrolysis. FESEM analysis revealed an increase in the specific surface area of the samples during fermentation, which may promote browning reactions [47]. With COC decomposition, the a* values (representing green or red color) increased significantly. *B. tropicus* can destroy the structure of cellulose and also produce pigments. The water-soluble pigments can change the color of the medium solution and the COC (Figure 7). The b* values (representing blue or yellow) are much lower than those before degradation. This change is probably due to the partial removal of proteins and soluble carbohydrates, substrates of the Maillard reaction [43]. Zheng et al. [47] similarly observed a color change in COC after treatment with cellulase.

#### 3.4.5. Bioethanol Production

The use of agricultural residues and waste materials to produce biofuel is attractive due to its advantages of renewability, non-competitiveness with food crops, reduction of waste, and lower greenhouse gas emissions [12,47]. Filtered crude enzymes (containing cellulase) produced by the hydrolysis of COC by *B. tropicus* released reducing sugars (Figure 8), reaching a maximum yield of 0.89 mg/mL (48 h).

*Saccharomyces cerevisiae* can be used as a fermenting agent to convert the sugar released by saccharification into bioethanol. Inoculating *Saccharomyces cerevisiae* at a low cell concentration in the early stage of fermentation will inhibit bioethanol production because yeast increases biomass by reproduction instead of producing bioethanol in the presence of a high reducing sugar concentration. Therefore, a high inoculum of yeast cells is required for ethanol production, so a 10% inoculum was selected here [3]. The trend of bioethanol production is the same as that observed in the estimation of reducing sugar, with the highest ethanol yield of 4.2 g/L observed after co-culture for 48 h (Figure 8). This study confirmed that COC is a promising raw material to produce bioethanol from lignocellulose, and future work should determine the optimal technical conditions for ethanol production from COC. This study is the first report of efficient enzymatic saccharification by *B. tropicus* of lignocellulosic biomass and demonstrates that the microbial-driven release of reducing sugars can successfully convert COC waste to ethanol. Application of this microbial hydrolysis process can decrease the total cost of bioethanol production. The microbial hydrolysis process is a result of the synergistic action of complex enzymes, and the present study only focused on CMCase, while other related enzymes need to be further measured and investigated in depth. In addition, the optimal enzyme production conditions of the target strain and the technical conditions for fuel ethanol production from COC should be further investigated by orthogonal experiments.

## 4. Conclusions

We isolated *B. tropicus* as a promising cellulase producer and demonstrated that this isolate can efficiently use COC as the sole carbon source to secrete cellulase. The cellulolytic potential was evaluated using FESEM, FTIR, XRD, and Color). The bacterial isolate was applied to the hydrolysis of COC to release reducing sugars, and then bioethanol was produced from the resulting reducing sugars. *B. tropicus* may be able to degrade other types of agricultural waste, making it a promising microbe for biomass waste management and biofuel production.

## Figures and Tables

**Figure 1 microorganisms-12-00240-f001:**
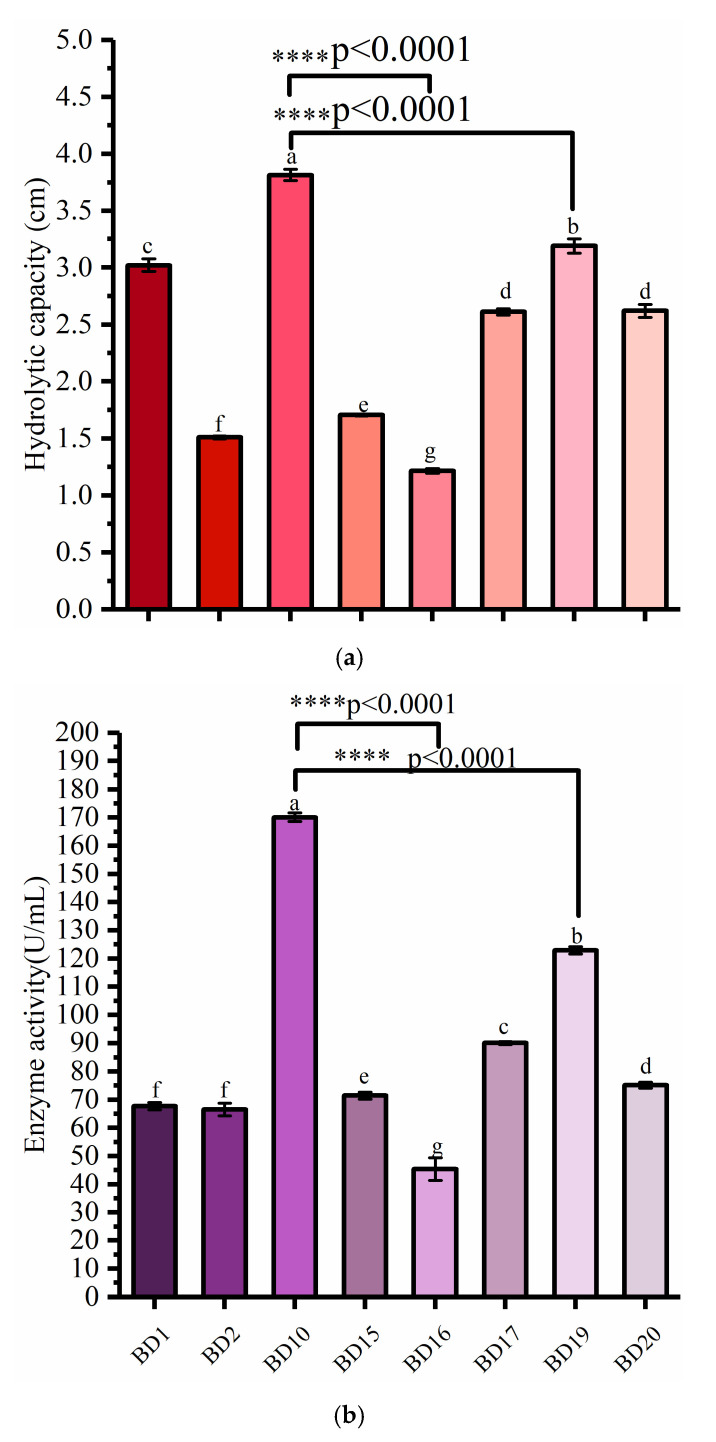
Activity comparison of screened strains (**a**) cellulase hydrolytic capacity. (**b**) cellulase enzyme activity. Different letters indicate significant difference (*p* < 0.05) between groups.

**Figure 2 microorganisms-12-00240-f002:**
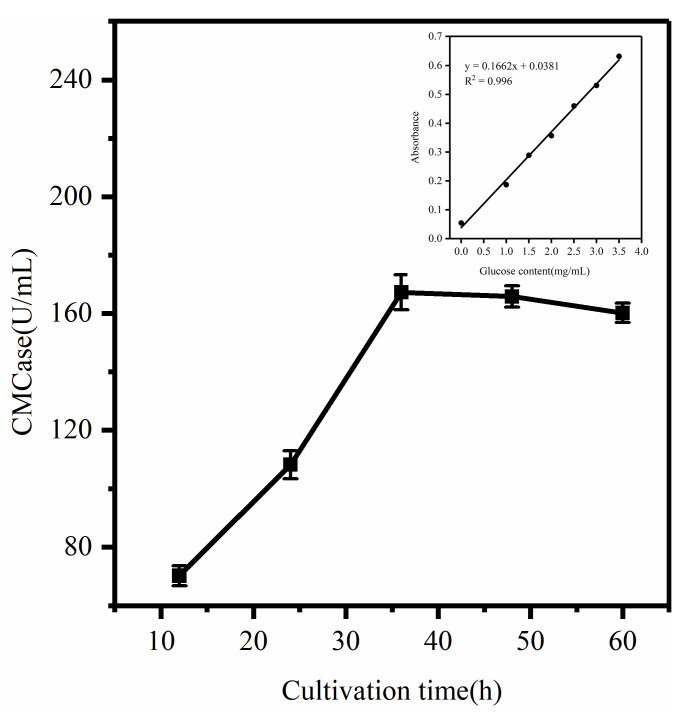
Determination of cellulase activity produced by bacteria degrading COC.

**Figure 3 microorganisms-12-00240-f003:**
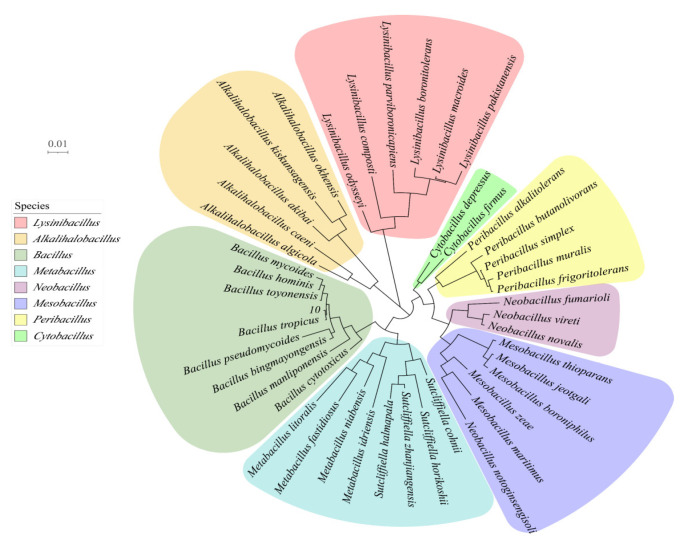
Phylogenetic tree of BD10.

**Figure 4 microorganisms-12-00240-f004:**
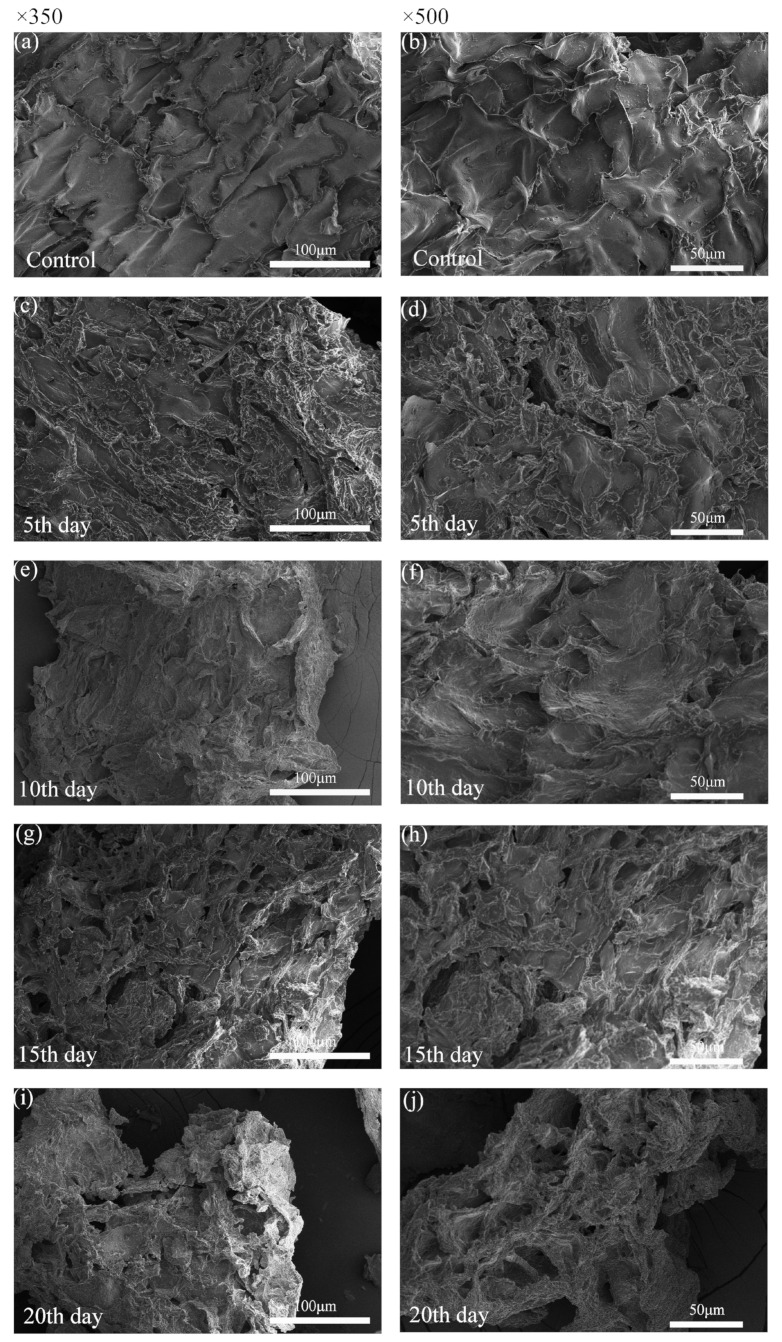
COC characterization by FESEM before and after treatment: (**a**,**b**) untreated; (**c**,**d**) treated for 5 days; (**e**,**f**) treated for 10 days; (**g**,**h**) treatment for 15 days; and (**i**,**j**) treatment for 20 days.

**Figure 5 microorganisms-12-00240-f005:**
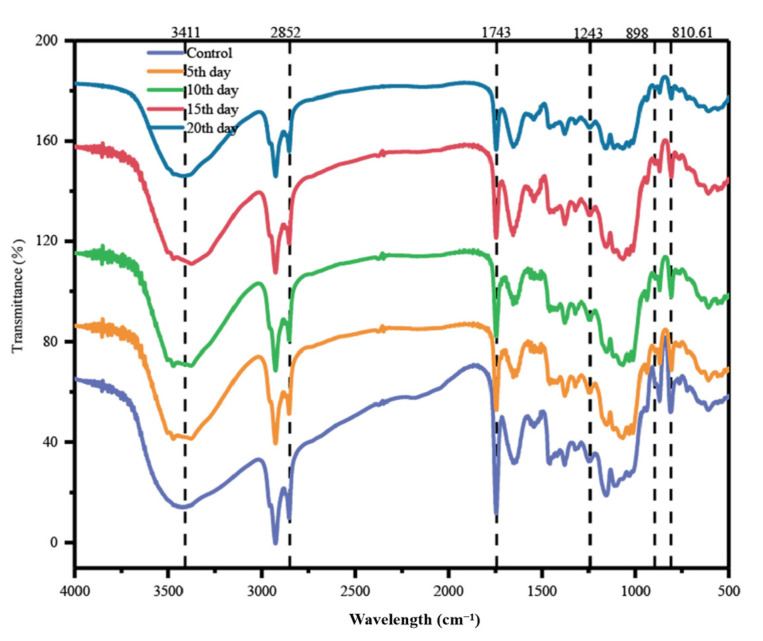
FTIR spectra of COC before and after treatment.

**Figure 6 microorganisms-12-00240-f006:**
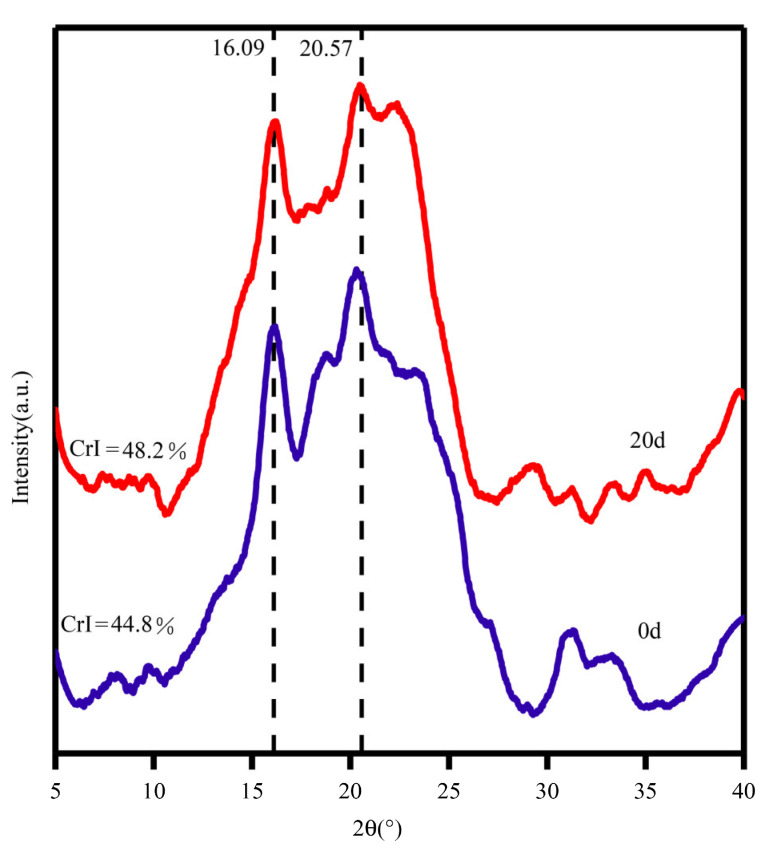
XRD spectra of COC before and after treatment by *B. tropicus*.

**Figure 7 microorganisms-12-00240-f007:**
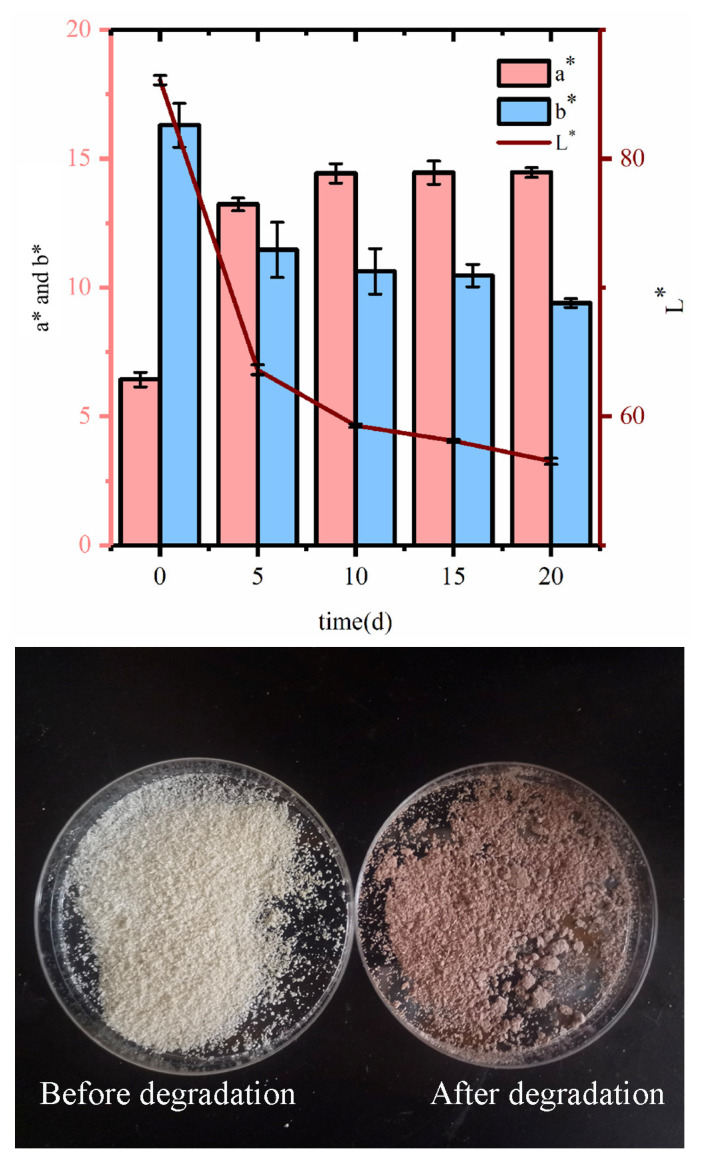
Color of COC before and after degradation.

**Figure 8 microorganisms-12-00240-f008:**
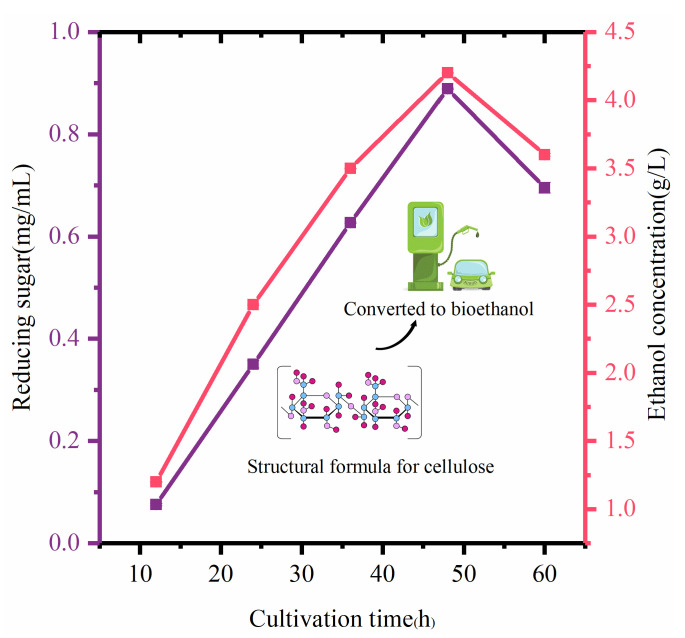
Conversion of COC to bioethanol.

**Table 1 microorganisms-12-00240-t001:** Comparison of maximum cellulase production of *B. tropicus* and other Bacillus strains.

Bacterial Strain	Isolated From	CMCase (U/mL)	Substrate	Pre-Treatment	Reference
*B. tropicus*	decayed dahlias	167.3	COC	NA	This study
*Bacillus* sp. PM06	Agro-waste Cocktail	0.150	Rice husk	NA	[33]
*Bacillus altitudinis* RSP75	The gut system of red flour beetle	47.1	Wheat husk	NA	[34]
*Bacillus* sp. JB1	Forest soil	5	Carboxymethylcellulose (CMC)	NA	[35]
*Bacillus subtilis* CNS	Forest soil	0.26	CMC	NA	[36]
*Bacillus Subtilis* Q3	Silage corn	18.667	CMC	NA	[37]
*Bacillus subtilis* MS 54	Paper and pulp industry waste	23.49	Maize bran	Sulfuric acid	[38]
*Bacillus pumilus* EB3	oil palm empty fruit bunch	0.079	CMC	NA	[39]
*Geobacillus stearothermophilus*	soil samples	1.94	sugarcane bagasse	NA	[40]

NA, not available.

## Data Availability

*B. tropicus* Sequencing data were submitted to the National Center for Biotechnology Information (NCBI) Sequence Read Archive (SRA) under Bio-Project:OP648255.

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
