# Peer review of "Identification of Cellulose-Degrading Bacteria and Assessment of Their Potential Value for the Production of Bioethanol from Coconut Oil Cake Waste"

_microorganisms, 2024, doi:10.3390/microorganisms12020240_

Round 1
Reviewer 1 Report
Comments and Suggestions for Authors
REVIEW MICROORGANISMS
Title – Identification of cellulose-degrading bacteria and assessment of their potential value for production of bioethanol from coconut oil cake waste
The manuscript deals with the isolation and identification of cellulose-degrading bacteria from decaying plants. The ability of an isolated bacteria to biodegrade COC was investigated as well as the ability to produce ethanol in co-cultivation with Saccharomyces cerevisiae. The English language is acceptable and the manuscript has the potential to be published by Microorganisms. However, it needs a major review before its acceptance. Below I made some comments (C), (Q) questions, or suggestions (S) to improve it.
Comments:
1. (S) – Abstract, pg.1, line 22 – It should be “....44.8% to 48.2% due to the hydrolysis of the amorphous region in...”.
2. (S) – Introduction, pg.2, line 49 – It should be “...on resources and their use, leading to an increase in total global energy...”.
3. (S) – Introduction, pg.2, line 67 – It should be “...low lysine concentration have led to a global decrease...”.
4. (S) – Introduction, pg.3, line 84 – It should be “...friendly method with a low yield of toxic substances...”.
5. (C) – Introduction – Authors must also comment on the role of lignin in the lignocellulosic biomass. Cellulose and hemicellulose were addressed but nothing was commented on lignin.
6. (S) – Materials and Methods, pg.3, line 115 – It should be “...bacteria. The method of Duhan et al. [39] was selected...”.
7. (S) – Materials and Methods, pg.3, line 119 – It should be “...agar media, incubated at 37oC for colony growth, ...”.
8. (C) – Materials and Methods, pg.4, line 137 – Please, cite the model and supplier for the used spectrophotometer.
9. (C) – Materials and Methods, pg.4, line 143 –It is necessary to define the CMCase activity as well as add the equation for its calculation.
10. (S) – Materials and Methods, pg.4, line 146 – It should be “...bacterial culture was inoculated into a culture medium with ...”. (Q) What was the inoculum concentration?
11. (S) – Materials and Methods, pg.4, line 164 – It should be “...tablets[26]. FTIR was performed with an experimental wavelength ...”.
12. (S) – Materials and Methods, pg.4, line 165 – It should be “...32 scans, and a resolution of ...”.
13. (S) – Materials and Methods, pg.5, line 184 – It should be “ A bacterial culture was used to inoculate a medium containing ...”.
14. (C) – Materials and Methods, pg.5, line 185 – Please, cite the model and supplier for the used shaker.
15. (C) – Materials and Methods, pg.5, line 186 – Please, cite the model and supplier for the used centrifuge.
16. (S) – Results and Discussion, pg.6, line 225 – It should be “ .....A bacterial culture was used to inoculate a medium containing ...”.
17. (C) – Results and Discussion, pg.6, line 234 – Please, add the Letters to show the significant difference among the strains in Figures 1 and 2. Additionally, despite agreeing with what the authors said regarding qualitative (on the plate) and quantitative analyses. Except for strains BD1 and BD20, there was a good correlation between the two strategies. Another point worth highlighting is that the authors quantified the activity of CMCase, that is, endocellulases. Perhaps it would be interesting to comment on this in the text.
18. (S) – Results and Discussion, pg.6, line 225 – It should be “.... production. Darwesh et al.[16] selected the fungus with ...”.
19. (S) – Results and Discussion, pg.7, line 254 – It should be “.... Darwesh et al.[35] reported that Aspergillus niger exhibited ...”. (C) Furthermore, I suggest that the authors compare it with another bacteria, rather than comparing it with a fungus.
20. (S) – Results and Discussion, pg.8, line 280 – It should be “.... differences may reflect variations in the cellulose ...”.
21. (S) – Results and Discussion, pg.8, line 281 – It should be “.... analysis conditions, or variations in the specificity ...”.
22. (C) – Results and Discussion, pg.8, line 282 – The reference Mrudula[42] was not identified in the article. Please, correct it.
23. (S) – Results and Discussion, pg.9, line 300 – It should be “.... caused significant morphological changes in the surface ...”.
24. (S) – Results and Discussion, pg.10, line 315 – It should be “.... of the fibrous fraction of the COC by cellulase, leading to...”.
25. (S) – Results and Discussion, pg.11, line 329 – It should be “.... of the fiber structure due to B.tropicus (Figure 5). In the data,...”.
26. (S) – Results and Discussion, pg.13, line 372 – It should be “During the fermentation period, the biological attack can cause a color change in the...”.
27. (C) – Results and Discussion, pg.15, line 410 – As previously mentioned, the authors only quantified the activities of endocellulases (CMCases). In general, when using lignocellulosic waste, microorganisms produce other enzymes such as exo-cellulases and beta-glucosidase. Therefore, considering the co-cultivation strategy used in the article, it becomes interesting. However, the amounts of ethanol produced are still very low. Thus, although the authors justify that the conditions are not optimized, I think they should comment that the cellulolytic extract produced by the bacteria could be used with already pre-treated biomass, that is, the emphasis would be on the hydrolysis stage, for later fermentation by the yeast S. cerevisiae.
28. (S) – Conclusion, pg.15, line 417 – It should be “..... degrade other types of agricultural waste, making it ...”.
References: Please, type the microorganism´s or plant´s name in italics. Refs. [2; 3; 5; 9; 15; 21; 24; 25; 27; 29; 30; 31; 32; 36; 38; 39; 41; 42; 43, 44; 46; 49; 51; 58; 59; 62; 63; 64; 66; 69].
Comments on the Quality of English Language
The English languague is acceptable.
Reviewer 2 Report
Comments and Suggestions for Authors
Fu et al. shows a highlight the research work conducted on the bioconversion of lignocellulosic biomass using Bacillus tropicus. The study was well characterized using FESEM, FTIR, XRD, color analysis and explore the potential application of B. tropicus, a co-culture of B. tropicus and Saccharomyces cerevisiae to produce ethanol from COC waste.
Overall, the paper is well written, and the results are well presented. The abstract, the introduction and literature are appropriate. The conclusion is in accordance with the results.
Only the Figure 3 It reads very badly; the drawing should be enlarged. And the reference is current.

Reviewer 3 Report
Comments and Suggestions for Authors
Identification of cellulose-degrading bacteria and assessment of their potential value for production of bioethanol from coconut oil cake waste
Manuscript by Zi-huan Fu et.al.
The review
Summary Statement:
Bacillus tropicus with cellulose degradation ability was isolated from decaying plants and used for coconut oil cake saccharification. Multiple methods incl. FTIR, electron microscopy, and X-ray diffraction were used to prove changes in the structure and properties of treated coconut oil cake. Cocultivation of B. tropicus and Saccharomyces cerevisiae on COC substrate resulted in 4.2 g/L ethanol production.
Major Strengths
The B.tropicus BD10 strain with relatively high CMC-ase activity was isolated and selected. The strain results in structural modification of plant cell wall and polysaccharides when cultured on COC. Also some reducing sugars are released during B.tropicus BD10 cultivation on COC.
Areas of Improvement
1. Low quality / very small symbols at some figures (eg. Fig. 1, Fig.2, Fig.3, Fig.7, and Fig.8). There is no interpretation of the designations in Figure 7.
2. Lines 276-277 The statement: “B.tropicus was not previously reported to be active for cellulose degradation” is not correct – see https://bacdive.dsmz.de/strain/140962 Section Metabolite utilization -> carboxymethylcellulose positive hydrolysis.
3. Lines 284-285 – High pathogenic potential of some B.tropicus strains was reported (https://doi.org/10.3390/pathogens12050693). It could limits the areas of application.
4. Most Bacillus cellulases hydrolyze synthetic carboxymethyl cellulose and soluble cellooligosaccharides, but barely hydrolyze rigid crystalline form of cellulose due to limited secretion of cellobiohydrolases [1-4] and this fact limits their application in lignocellulose bioconversion processes. However, cellulosome-containing anaerobic bacteria like Acetivibrio and Clostridium obtains full range of enzymes needed for efficient cellulose and hemicellulose degradation to simple sugars (see also Schwarz, W.H., Liebl, W., or/and Zverlov, V.V. publications). Authors clearly illustrated by different methods that only some part of COC is hydrolyzed/converted to simple sugars by B.tropicus. Therefore, the statement that the selected strain has a potential for bioethanol production it seems far-fetched and unsupported by real facts.
Table 1 should include a comparison of the selected strain BD10 with other cellulose-degrading microorganisms, not just Bacillus.
[1] Production and characterization of cellulase by Bacillus pumilus EB3
Int J Eng Technol, 3 (2006), pp. 47-53
[2] Avicelase production by a thermophilic Geobacillus stearothermophilus isolated from soil using sugarcane bagasse World Acad Sci Eng Technol, 57 (2009), pp. 487-491
[3] Bacillus pumilus S124A carboxymethyl cellulase; a thermostable enzyme with a wide substrate spectrum utility Int J Biol Macromol, 67 (2014), pp. 132-139, 10.1016/j.ijbiomac.2014.03.014
[4] Current Trends in Research and Application of Microbial Cellulases. Research Journal of Microbiology Vol 6 (1), 2011 41-53
5. It is difficult or even impossible to judge the novelty of the isolated strains based on the data presented in the manuscript. It is necessary to present data on the novelty and specific features of the strains obtained.
6. Lines 413-414 – “The cellulolytic potential was evaluated using FESEM, FTIR, XRD, and chromatography” – the use of chromatography could be hardly found in the manuscript.
7. The use of K2Cr2O7 reagent for ethanol production evaluation is very controversial due to the low selectivity of the determination. GC or HPLC analysis for ethanol quantification need to be used.
8. Line 393 – The yield of reducing sugar increased to 0.89 mg/mL from 20 g/L of COC, indicating less than 5% substrate conversion. It is evident that the enzymatic conversion of COC lignocellulose did not yield optimal results. However, 4.2 mg/mL of ethanol was observed (Line 402) - the mass balance is clearly out of sync.
9. Lines 375-383 Maillard reaction typically occurs at higher temperatures. The brownish colour of COC treated by B.tropicus is most likely due to oxidation of phenolic compounds or formation of pigments rather than melanoidins.
10. Line 18 - …cellulose after treatment with COC?
11. Line 247 - …relatively stable cellulase (not cellulose)?
12. Line 507, Ref 32. Please change the author list: GÄ™sicka, A., Borkowska, M., BiaÅ‚as, W, Kaczmarek, P., CeliÅ„ska, E. …
13. Minor spelling issues.
Conclusion:
The manuscript could be published only after deep and thorough modification and additional experiments.
Although the BD10 strain and other BD strains exhibit relatively high CMC activity, they are not suitable candidates for second-generation bioethanol production due to their inability to deeply degrade lignocellulose from coconut oil production wastes. It may be worth considering alternative applications for these strains, but bioethanol production is not recommended.
Comments on the Quality of English LanguageEditing and proofreading of the title, text, and the reference list, as well as revision of figures, are necessary.
Reviewer 4 Report
Comments and Suggestions for Authors
Manuscript of Liu et al. concerns isolation of bacteria (Bacillus tropicus), using this strand to the degradation of agricultural waste and investigation of this process for ethanol production. I think that the subject of the work will be interesting to readership of Microorganisms journal and in general to biologists, ecologists, chemists etc. working in different fields of science. The conclusions may be regarded as reliable due to using modern physical methods (electron microscopy, XRD, IR spectroscope etc.) for investigation. Although the text was written significantly interesting, I found several inconsistences (like very strange recording of Chinese names of Authors). Moreover, the chemical sense of the data are presented poor; as a main Note I recommend to perform Major Revision with using Schemes of chemical reactions used.
Additional Comments:
1) Abstract requires significant editing correction (“… isolated and screened (?) from decaying plants”; “… electron microscopy… showed … efficiency (?) of hydrolysis”).
2) Line 77: “hemicellulose”. What is it?
3) Line 117: “about 1 cm…” I don’t understand how it is possible to use these data or reproduce them for checking.
4) Line 141: why 13 min?
5) L. 206: what means “hydrolysis capacities”?
6) Fig. 2: why there a max on a curve? Why there is decreasing after achieving the max value?
7) L. 309: “… toxic… organic acids”. Which ones? Without reaction Schemes it’s hardly to understand this text!
8) For polymers used add special characteristics (PDI, Mw etc.).
9) L. 393: “… reducing sugars”. Which ones? How they were established?
10) Fig. 8: the structure is too abstract. It requires retyping.
11) L. 414 (conclusion): “… chromatography”?..
12) Reference 32: use Last Names for Authors.
Comments on the Quality of English LanguageModerate editing of English language required.
Round 2
Reviewer 4 Report
Comments and Suggestions for Authors
The Manuscript was corrected. I believe, that it is ready for publication.